**Filologia e Wikidata. Per una riorganizzazione del lessico delle Edizioni Critiche**

Il lessico della filologia, in particolare quello legato alle edizioni critiche, trova su Wikidata una rappresentazione scarsa e frammentaria. Gli item relativi ai concetti fondamentali di questo ambito sono pochi, spesso sommari, difficili da reperire tramite query SPARQL e privi delle sfumature concettuali e del dibattito intellettuale che la filologia ha sviluppato nel corso del tempo. Questa lacuna compromette l'efficacia di Wikidata come strumento per la ricerca accademica e la condivisione interdisciplinare della conoscenza.

La proposta mira a riorganizzare e strutturare il lessico delle edizioni critiche utilizzando sia **Wikidata** sia **Wikibase**, la piattaforma open source alla base di Wikidata. Wikibase consentirà di creare un ambiente controllato e personalizzabile per la modellazione e il test del lessico, garantendo una maggiore flessibilità nell'organizzazione dei dati e facilitando un eventuale trasferimento su Wikidata.

Il progetto utilizza come punto di partenza il **LexiconSe (Lexicon of Scholarly Editing)**, una risorsa collaborativa, aperta e multilingue che raccoglie definizioni provenienti da articoli, monografie e altre fonti accademiche sulle edizioni critiche e sulla filologia in senso lato. LexiconSe sarà integrato in una istanza di Wikibase, dove i termini saranno organizzati prendendo come punto di riferimento ontologie consolidate come la **Critical Apparatus Ontology (CAO)** e la **Scholarly Editing Ontology**.

La proposta intende coinvolgere la comunità accademica, la comunità Wikidata e Wikimedia, in senso più ampio, favorendo un approccio collaborativo che porti a riflettere sulla necessità di elaborare linee guida per i contributori relativamente alle tematiche della filologia.