# OpenReview forum: "Filologia e Wikidata. Per una riorganizzazione del lessico delle Edizioni critiche"
_wikimedia.it/Wikidata_and_Research/2025/Conference — WD&R Poster_

### Official Review · ~Lucia_Sardo1 · 2025-01-07
**revisione**

**Originality:** 5
**Impact:** 5
**Confidence:** 4

**Review:**

La proposta presentata risulta abbastanza ben strutturata e con obiettivi chiaramente definiti; l'autore, partendo da una buona conoscenza degli strumenti e delle risorse esistenti nell'ambito disciplinare di interesse, propone un interessante utilizzo di Wikibase per la strutturazione dei dati relativi al lessico delle edizioni critiche integrando fonti già esistenti. La proposta può essere di sicuro interesse per gli studiosi della disciplina e per coloro che sono interessati a lavorare sugli aspetti lessicologici.

**Compliance:**

4

**Scientific Quality:**

4

---

### Official Review · ~Monica_Berti1 · 2025-01-10
**Un contributo importante per una discussione sul coinvolgimento dei filologi in Wikidata**

**Originality:** 5
**Impact:** 5
**Confidence:** 5

**Review:**

L'autore presenta una proposta per riorganizzare e strutturare il lessico delle edizioni critiche utilizzando Wikidata e Wikibase. L'intento è sicuramente molto ambizioso e il progetto parte con l'utilizzo della risorsa collaborative Lexicon of Scholarly Editing (LexiconSe). La proposta è interessante come stimolo al coinvolgimento della comunità accademica in Wikidata e Wikimedia per favorire un approccio collaborativo nell'ambito della filologia, che ancora necessita di una spinta in questa direzione.

**Compliance:**

4

**Scientific Quality:**

4

---

### Decision · Program_Chairs · 2025-02-05

**Decision:**

Withdrawn

**Comment:**

Ritirato.
==New format: Poster==
presence confirmed

Dear Author,
thank you very much for your proposal. We regret to inform you that your proposal was not selected among the papers.

Even if not selected as paper, we consider your proposal relevant and interesting and we would like to propose you to prepare instead a lightening talk (if you - or another member of your team - can participate in presence at the conference) or a poster (which can be exhibited even if you will not attend the conference).

It would be a pleasure to learn more about your work through a lightening talk or a poster.
Thank you for submitting a proposal and please let us know if you like the idea of converting it into a lightening talk or a poster and which format you prefer.

Regards,
The scientific committee of the conference Wikidata and Research